# LONGVITU: INSTRUCTION TUNING FOR LONG-FORM VIDEO UNDERSTANDING

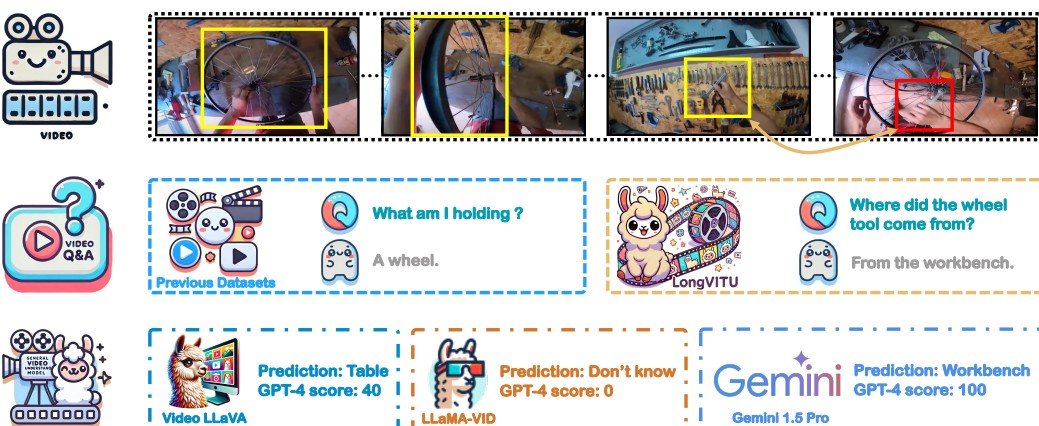

Figure 1: **Examples of LongViTU.** The top row shows the video sequence, with the `yellow box` indicating the answer and the `red box` highlighting relevant objects. The middle row presents a video Q&A example, showing that LongViTU captures fine-grained spatial details and temporal reasoning, whereas previous datasets focused on static spatial features. The bottom row shows predictions from canonical video understanding models, evaluated by GPT-4 against ground truth based on a novel predefined scoring criteria we designed.

## ABSTRACT

This paper presents LongViTU, a large-scale (~121k QA pairs, ~900h videos), automatically generated dataset for long-form video understanding. Our key idea is inspired by the success of Large Language Models (LLMs) and Multimodal Language Models (MLMs) that are fueled by machine-generated instruction-following data (*e.g.*, InstructGPT, LLaVA). We developed a *systematic* approach to produce massive question-answeringing pairs tailored to virtually unbounded long videos by organizing them into a **hierarchical tree**, incorporating **self-revision** mechanisms to guarantee high quality. We curate LongViTU for each QA pair: 1) involves a long context (average *certificate length* of 4.6 minutes); 2) requires rich knowledge and condensed reasoning (commonsense, causality, planning, *etc.*); 3) explicit labels the timestamps of relevant events throughout the entire video. Furthermore, LongViTU provides a benchmark to facilitate future research in instruction-following for long-form videos. Our experiments first reveal the performance gap between open-source video MLMs and their commercial counterparts (*e.g.*, Gemini-1.5-Pro) on this benchmark. Supervised Fine-Tuning (SFT) on open-source models led to Video-LLaVA achieving the best performance, with a GPT-4 score of 50.7, closely following 52.3 by the leading closed-source model Gemini-1.5-Pro, underscoring the substantial challenge posed by our benchmark. Further SFT on LongViTU with Video-LLaVA resulted in improvements of 30.7% on the In-Distribution (ID) benchmark EgoSchema; 12.9% and 0.6% on the Out-of-Distribution (OOD) benchmarks WorldQA and VideoMME, respectively. These outcomes demonstrate the effectiveness and robust OOD generalizability of our proposed instruction-tuning scheme for long-form video understanding. The dataset, SFT models, and code are publicly available on the anonymous page LongViTU.

# 1 INTRODUCTION

We introduce LongViTU, a novel dataset tailored for large-scale, long-form video understanding (see Figure 1 for examples, more in Table 4). In comparison to existing automatically generated Video question-answering (VQA) datasets, such as Otter (Li et al., 2023a), Video-ChatGPT (Maaz et al., 2023), InternVideo (Wang et al., 2022), VideoChat (Li et al., 2023c), MVBench (Li et al., 2024), EgoSchema (Mangalam et al., 2024) and CinePile (Rawal et al., 2024), *etc.*, LongViTU incorporates key advancements that yield a dataset of greater naturalness and diversity. We delineate the primary advantages of our dataset in contrast to the limitations of prior works downsides below, see Table 1 for a clearer view.

- **Diverse real world scenarios.** Some prior VQA datasets originate from videos captured in virtual environments, such as Env-QA (Gao et al., 2021) and OpenEQA (Majumdar et al., 2024), which inherently introduce a domain gap. Many other datasets, despite utilizing real world videos, often feature limited or homogeneous scenes. For instance, EgoVQA (Fan, 2019) predominantly includes office scenes, EgoTaskQA (Jia et al., 2022) primarily focuses on home environments, EgoSchema (Mangalam et al., 2024) encompasses a very limited number of scenes, WorldQA (Zhang et al., 2024b) is mainly based on YouTube short videos, and both MovieChat (Song et al., 2024) and CinePile (Rawal et al., 2024) are derived from movie videos. In contrast, LongViTU leverages the complete Ego4D (Grauman et al., 2022), this extensive dataset enables VQA tasks to encompass a broad spectrum of real world scenarios.

- **Explicit timestamp labels.** Previous datasets lack explicit timestamp labels for QA-related events, meaning that while a video may contain multiple QAs, the precise start and end times for each QA are not provided. Our hierarchical pipeline organizes video content into a tree structure, enabling QA generation on subtrees and thereby ensuring explicit, accessible timestamps for each event. Consequently, LongViTU offers precise start and end timestamps for all QA events, supporting accurate identification of key events within ultra-long, redundant video sequences, and enabling comprehensive model performance analysis.

- **Long certificate length.** The average *certificate length* (introduced in EgoSchema (Mangalam et al., 2024), which we adhere to) in most short VQA datasets, such as NextQA (Xiao et al., 2021) and ActivityNet-QA (Yu et al., 2019), is typically under 10 seconds, despite the total video duration spanning tens or hundreds of seconds. Some longer datasets, like WorldQA (Zhang et al., 2024b), still feature an average certificate length of less than 60 seconds, while EgoSchema (Mangalam et al., 2024) remains below 100 seconds. In contrast, our approach supports the generation of QAs across a spectrum of durations, from brief to notably extended sequences. As a result, LongViTU achieving an average certificate length of 276.8 seconds (~4.6 minutes), encompassing a diverse temporal range from events as brief as 6 seconds to those exceeding 1 hour. For further statistical details, please refer to Figure 3.

- **Fine-grained categorization.** Existing VQA datasets often lack comprehensive categorization, primarily focusing on basic question types that revolve around spatial elements such as objects, attributes, locations, and states, *etc*. Although recent efforts like EgoTaskQA (Jia et al., 2022) and OpenEQA (Majumdar et al., 2024) have introduced categorized questions, these classifications remain relatively coarse. In contrast, LongViTU places greater emphasis on the spatial-temporal interplay, providing fine-grained categorization by incorporating detailed spatiotemporal features intrinsic to video contexts. This includes three primary categories: Spatiotemporal Understanding, Episodic Reasoning, and Commonsense Inference, as well as more fine-grained categories, as shown in Figure 3b and Table 4.

- **Open-ended precise QA.** As opposed to the multiple-choice formats employed in VQA datasets such as EgoVQA (Fan, 2019), EgoSchema (Mangalam et al., 2024), and CinePile (Rawal et al., 2024), or despite the open-ended formats in MovieChat (Song et al., 2024), WorldQA (Zhang et al., 2024b), and OpenEQA (Majumdar et al., 2024), which frequently feature irrelevant or redundant answers. LongViTU ensures a closer alignment between questions and video content, with answers being succinct and directly relevant. This is achieved through a self-revision mechanism that refines the QA by removing redundancies and further aligning questions with the video content, thereby ensuring concise, relevant, and high-quality QAs, which is detailed in Appendix B.

By preserving these advantages during dataset construction, as a result, the final dataset LongViTU comprises ~121k high-quality QA pairs within ~900 hours of videos across 3 primary with 12 fine-grained categories (detailed in Table 4). To the best of our knowledge, LongViTU is the first publicly available long-form video question-answering dataset featuring explicit QA-related timestamp

Table 1: **Comparison with previous datasets.** The video sources for each dataset are listed under "Base", where "N/A" indicates that videos are sourced from a collection of movies without a specific origin. Furthermore, * denotes multiple-choice answers, while ** indicates open-ended answers, LongViTU is the first large-scale dataset designed for long-form video understanding with explicit timestamp labels. The video durations and the number of QA pairs are approximate.

| Dataset | Base | Scenario | Open-ended Answer | Fine-grained Categorization | Explicit Timestamp | Video Duration | QAs |
|---|---|---|---|---|---|---|---|
| EgoVQA | IU Multiview (Xu et al., 2018) | real world | ✗* | ✗ | ✗ | 10h | 600 |
| Env-QA | AI2-THOR (Kolve et al., 2017) | virtual env | ✓** | ✗ | ✗ | 130h | 85.1K |
| EgoTaskQA | LEMMA (Jia et al., 2020) | real world | ✓ | ✓ | ✗ | 15h | 40K |
| EgoSchema | Ego4D (Grauman et al., 2022) | real world | ✗ | ✗ | ✗ | 250h | 5K |
| MovieChat | N/A | movie | ✓ | ✗ | ✗ | 160h | 13K |
| WorldQA | PVSG (Yang et al., 2023b) | real world | ✓ | ✗ | ✗ | 10h | 1K |
| CinePile | N/A | movie | ✗ | ✗ | ✗ | 420h | 303K |
| OpenEQA | ScanNet (Dai et al., 2017) HM3D (Ramakrishnan et al., 2021) | virtual env | ✓ | ✗ | ✗ | 3h | 1.6K |
| **LongViTU (ours)** | Ego4D (Grauman et al., 2022) | real world | ✓ | ✓ | ✓ | 900h | 121K |

annotations, constructed through a hierarchical pipeline and incorporating self-revision mechanisms. In summary, our contributions are as follows:

- We propose a novel automatic pipeline to generate Video question-answering data, mitigating several limitations of existing datasets: diverse real world scenarios, explicit timestamp labels, long certificate length, fine-grained categorization, and open-ended precise QA.
- With our pipeline, we curate LongViTU, a large-scale high-quality dataset and benchmark aimed at advancing instruction tuning for long-form video understanding.
- We conducted extensive experiments demonstrating the benefits of LongViTU for canonical Vision Language Models and providing insights into the critical design principles underlying our approach.

## 2 THE LONGVITU DATASET

We developed a hierarchical approach to process indefinitely long-form video content by organizing it into a tree structure, enabling the generation of appropriate QA while capturing detailed spatial information from individual frames and the temporal relationships between events or objects across varying duration scales. This framework facilitates the generation of QA pairs with explicit timestamps and long certificate length, enabling fine-grained categorization that aligns with the video content, and also provides open-ended precise QA. To the best of our knowledge, no existing automated Video question-answering dataset generation methods offer these capabilities.

### 2.1 DATASET PIPELINE

#### 2.1.1 STAGE I: HIERARCHICAL VIDEO TREE CONSTRUCTION

**Frame level.** Commencing at the *frame level*, we employ InternLM-XComposer2 (Dong et al., 2024) to perform multi-frame dense captioning (sampled at 1 fps) across annotated events in the Ego4D (Grauman et al., 2022). The descriptions of video context at this stage are denoted by $\langle d_f, t_s^f, t_e^f \rangle$, where $t_s^f$ and $t_e^f$ represent the respective start and end times. Accurate timestamps are derived from Ego4D's temporal annotations for each event provided by human annotators.

**Event level.** Redundant text frequently emerges in the *frame level*, to mitigate this issue, we employ GPT-4 (Achiam et al., 2023) for processing both manually annotated events $d_f^H$ from the original Ego4D (Grauman et al., 2022), which offer precise temporal contexts, and automatically generated dense captions $\langle d_f^1, ..., d_f^F \rangle$, to eliminate redundancy and refine the descriptions at the *event level*. GPT-4 then restructures these annotations into succinct *event level* descriptions, represented as $\langle d_e, t_s^e, t_e^e \rangle$, where $t_s^e$ and $t_e^e$ denote the start and end times of the events, respectively.

**Segment level.** Thereafter, utilizing GPT-4 organizes events into segments within the hierarchical video tree $\mathcal{T}_{\text{video}}$, where closely related consecutive events are merged to form segments, subsequently summarizing these into *segment level* descriptions, denoted as $\langle d_s, t_s^s, t_e^s \rangle$. Consequently, the video content is structured hierarchically with the tree root, segments and events as intermediate nodes, and frames as leaf nodes.

**Video Tree formulation.** Drawing on the discussion above, we formalize the hierarchical tree structure for long-form video content as follows:

Figure 2: **Pipeline of LongViTU.** We adopt a ***hierarchical*** pipeline that organizes video content into a tree structure, with subtrees encapsulating information at different temporal scales. This framework facilitates the generation of QA pairs with *explicit timestamps*, ensuring adaptability through varying *contextual lengths*. Furthermore, by summarizing content across multiple temporal levels (*frame level, event level, segment level*), our approach enables LLMs to generate distinct types of questions, resulting in a *fine-grained categorization* aligned with the video content. Finally, a ***self-revision*** step eliminates redundancy and prior information, thereby enhancing the overall quality of LongViTU. For more details, please refer to Section 2.

$$\mathcal{T}_{\text{video}} = \left\{ \langle R, \{\langle d_s^i, t_s^{s^i}, t_e^{s^i}, \{\langle d_e^j, t_s^{e^j}, t_e^{e^j}, \{\langle d_f^k, t_s^{f^k}, t_e^{f^k}\rangle\}_{f=1}^F \rangle\}_{e=1}^E \rangle\}_{s=1}^S \rangle \right\} \tag{1}$$

where $\mathcal{T}_{\text{video}}$ represents the hierarchical tree structure of the video, with $R$ as the root node, and $\langle d_s^i, t_s^{s^i}, t_e^{s^i}\rangle$ denoting *segment level* descriptions (children of the root). Each segment contains multiple *event level* descriptions $\langle d_e^j, t_s^{e^j}, t_e^{e^j}\rangle$, while each event contains multiple *frame level* descriptions $\langle d_f^k, t_s^{f^k}, t_e^{f^k}\rangle$. Here, $S$, $E$, and $F$ represent the total number of segments, events, and frames, respectively.

### 2.1.2 STAGE II: LONG-FORM QA GENERATION

**Sliding window.** The application of a *sliding window* approach to any subtree of $\mathcal{T}_{\text{video}}$ enables the generation of QAs that adeptly capture both the spatial intricacies of individual frames and the temporal continuity among segments. In this implementation, the sliding window encompasses five segments sequentially, integrating descriptions at both the segment and event levels. This method ensures the capture of continuous events (*long-term temporal relevance*) and detailed spatial features (*short-term spatial relevance*). To prevent premature generation of questions concerning recent events, GPT-4 is programmed to formulate questions based on critical events identified within the initial three segments, while deriving answers from the following two segments.

**Crucial advantages.** This implementation synergizes with the hierarchical structure of the video tree, yielding several crucial advantages that enhance the efficacy of the LongViTU methodology:

- Explicit timestamp labels: each tree node is clearly marked with event timestamp, thereby improving the precision of temporal analysis.
- Long certificate length: the capability to engage with various subtrees permits the handling of QAs across a broad spectrum of durations, ranging from brief to extended periods, thus facilitating versatile management of content length.
- Fine-grained categorization: Focusing on specific subtrees significantly reduces the input text for the LLM, enabling it to handle long temporal events while maintaining attention to rich spatial details. This approach enables generation of QAs across diverse, fine-grained categories

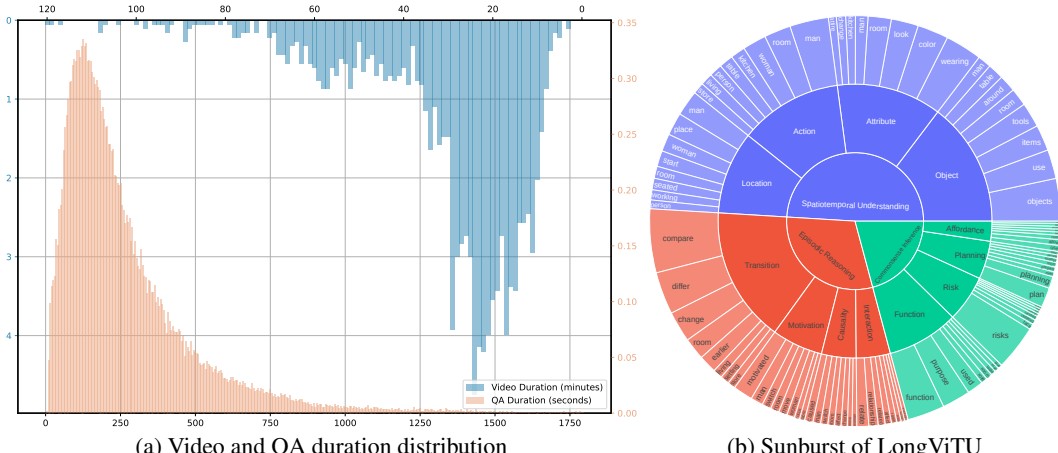

(a) Video and QA duration distribution      (b) Sunburst of LongViTU

Figure 3: **Statistics of LongViTU.** Subfigure (a) depicts the distributions of video and QA durations within LongViTU. The bottom horizontal axis (from left to right) represents QA duration in seconds, while the right vertical axis (from bottom to top) shows the percentage of the total dataset. QAs predominantly vary from 6 to 500 seconds, with an average duration of 276.8 seconds. The top horizontal axis (from right to left) details the video durations, and the left vertical axis (from top to bottom) presents the LongViTU's percentage distribution. Video lengths mostly range from 5 to 40 minutes, averaging 29.3 minutes, and follow a long-tail distribution. Subfigure (b) illustrates the QA categorization in LongViTU along with their word frequency distribution. The outermost ring of the sunburst chart displays the eight most frequent words within each category, with segment sizes reflecting their proportional frequency within LongViTU. Zoom in for a better view.

**QA generation.** We generate the QAs from the subtree descriptions within the sliding window, formalized as:

$$\mathcal{QA}_{\text{video}} = \left\{ \left\langle Q, A, \left\{ \left\langle d_s^i, \left\{ \left\langle d_e^j, \left\langle d_f^k \right\rangle \right\}_{f=1}^F \right\rangle \right\}_{e=1}^E \right\}_{s=1}^5 \right\rangle \right\}$$ (2)

where $\mathcal{QA}_{\text{video}}$ denotes the generated QA process applied on the $\mathcal{T}_{\text{video}}$. The notation $\{d_s^i\}$ represents the five segments chosen through the sliding window technique. Each segment encapsulates event level descriptions $\{d_e^j\}$ and frame level descriptions $\{d_f^k\}$. The model GPT-4 is tasked with formulating questions from notable events within the first three segments, while deriving answers from the remaining two. The parameters $S$, $E$, and $F$ signify the total counts of segments, events, and frames, respectively, with $Q$ and $A$ representing the question-answering pairs generated.

### 2.1.3 STAGE III: SELF-REVISION

**Self-Revision.** In this stage, GPT-4 conducts a thorough review of the generated question-answering pairs. This self-revision is pivotal for maintaining quality of QA pairs. The GPT-4 reviews each event description associated with the question-answering pairs to verify their consistency with the underlying video context. It identifies and rectifies any deviations or fabricated elements, extraneous information is minimized to highlight critical aspects of the question-answeringing, thereby preventing the inclusion of redundant or overly simplistic responses. Further details regarding the prompts and human evaluations of this self-revision process are detailed in Appendix B.

### 2.2 CHARACTERISTICS AND STATISTICS

**Duration distribution.** The LongViTU dataset comprises 1,833 videos, split into 1,533 for training, 200 for validation, and 100 for testing, totaling ~900 hours. The average video duration is 29.3 minutes, ranging from 3.5 to 120.7 minutes with a standard deviation of 17.5 minutes, and follows a long-tail distribution (refer to Figure 3a ). QA durations vary between 6 and 1800 seconds, with an average of 276.8 seconds and a standard deviation of 257.9 seconds, also showing a long-tail pattern. The average durations of events and segments are 8.5 and 82 seconds, respectively. In total, the LongViTU dataset includes 121k QA pairs: 101k for training, 14k for validation, and 6k for testing, which also serve as a benchmark.

**Frequency distribution.** The sunburst diagram of LongViTU is illustrated in Figure 3b, question-answeringing pairs are categorized into three primary groups: **Spatiotemporal Understanding (55%)**, sub-divided into Object (12.2%), Attribute (10.7%), Location (15.5%), Action (16.6%);

Table 2: **Quantitative results on LongViTU.** All results are derived from evaluations conducted by GPT-4 (Achiam et al., 2023), the criteria and prompt are detailed in Appendix B. * denotes results obtained in a zero-shot manner, while ** indicates fine-tuned results following training on the LongViTU training set, △ compared highlighting the percentage difference in performance between their. Overall Avg. represents the average scores across three primary categories. The top-performing open-source model, Video-LLaVA (Lin et al., 2023), achieved a score of 50.7, approaching the 52.3 score of the best commercial model, Gemini-1.5-pro (Reid et al., 2024). Frame-based models uniformly sampled 8 frames from videos, while sampling-based models captured 1 frame per second (1 fps), with the exception of VideoAgent (Fan et al., 2024), which processes 1 frame every 2 seconds (1/2 fps).

| Setting | Method | Overall Avg. | Spatiotemporal Understanding | | | | |
|---|---|---|---|---|---|---|---|
| | | | Object | Attribute | Location | Action | Avg. |
| **Blind** | GPT-4 turbo | 38.2 | 26.1 | 33.2 | 32.0 | 29.4 | 30.2 |
| **Frame-Based** | mPLUG-OWL* | 42.4 | 33.5 | 37.6 | 43.6 | 35.4 | 37.8 |
| | Video-LLaVA* | 45.9 | 37.8 | 46.3 | 49.1 | 38.1 | 42.7 |
| | Video-LLaVA** | 50.7 | 39.3 | 49.2 | 49.6 | 41.8 | 44.9 |
| | △ compared | +10.5% | +4.0% | +6.3% | +1.0% | +9.7% | +5.2% |
| **Sampling-Based** | VideoAgent* | 44.0 | 35.7 | 43.1 | 45.9 | 36.4 | 40.2 |
| | LLaMA-VID* | 38.2 | 29.4 | 35.6 | 40.1 | 31.5 | 34.3 |
| | LLaMA-VID** | 44.5 | 33.5 | 37.4 | 45.7 | 37.6 | 39.1 |
| | △ compared | +16.5% | +13.9% | +5.1% | +14.0% | +19.4% | +14.0% |
| | Gemini-1.5-Pro* | 52.3 | 54.3 | 58.6 | 56.3 | 48.1 | 54.7 |

| Setting | Method | Overall Avg. | Episodic Reasoning | | | | |
|---|---|---|---|---|---|---|---|
| | | | Transition | Interaction | Causality | Motivation | Avg. |
| **Blind** | GPT-4 turbo | 38.2 | 45.1 | 47.4 | 47.7 | 56.1 | 49.5 |
| **Frame-Based** | mPLUG-OWL* | 42.4 | 45.8 | 47.7 | 47.7 | 49.4 | 47.6 |
| | Video-LLaVA* | 45.9 | 45.6 | 50.5 | 48.8 | 53.2 | 49.4 |
| | Video-LLaVA** | 50.7 | 50.5 | 56.4 | 59.7 | 64.9 | 58.0 |
| | △ compared | +10.5% | +10.7% | +11.7% | +22.3% | +22.0% | +17.4% |
| **Sampling-Based** | VideoAgent* | 44.0 | 43.1 | 45.5 | 49.9 | 52.8 | 48.1 |
| | LLaMA-VID* | 38.2 | 40.4 | 46.7 | 40.5 | 46.6 | 43.2 |
| | LLaMA-VID** | 44.5 | 46.7 | 48.4 | 54.2 | 57.7 | 52.1 |
| | △ compared | +16.5% | +15.6% | +3.6% | +33.8% | +23.8% | +20.6% |
| | Gemini-1.5-Pro* | 52.3 | 47.8 | 45.5 | 47.8 | 47.5 | 47.3 |

| Setting | Method | Overall Avg. | Commonsense Inference | | | | |
|---|---|---|---|---|---|---|---|
| | | | Planning | Risk | Function | Affordance | Avg. |
| **Blind** | GPT-4 turbo | 38.2 | 36.5 | 51.1 | 55.9 | 50.9 | 48.7 |
| **Frame-Based** | mPLUG-OWL* | 42.4 | 42.1 | 54.6 | 54.3 | 51.5 | 50.3 |
| | Video-LLaVA* | 45.9 | 41.6 | 56.8 | 55.3 | 54.6 | 51.7 |
| | Video-LLaVA** | 50.7 | 50.2 | 62.6 | 64.0 | 64.6 | 59.8 |
| | △ compared | +10.5% | +20.7% | +10.2% | +15.7% | +18.3% | +15.7% |
| **Sampling-Based** | VideoAgent* | 44.0 | 40.0 | 53.7 | 55.5 | 53.1 | 50.7 |
| | LLaMA-VID* | 38.2 | 34.9 | 51.3 | 46.5 | 47.2 | 44.1 |
| | LLaMA-VID** | 44.5 | 43.9 | 54.5 | 55.7 | 53.8 | 51.7 |
| | △ compared | +16.5% | +25.8% | +6.2% | +19.8% | +14.0% | +17.2% |
| | Gemini-1.5-Pro* | 52.3 | 43.6 | 57.5 | 46.1 | 43.6 | 50.3 |

**Episodic Reasoning (24.4%)**, including Transition (8.1%), Interaction (3.4%), Causality (5.4%), Motivation (7.5%); and **Commonsense Inference (20.6%)**, composed of Planning (5.4%), Risk (2.7%), Function (6.4%), and Affordance (4.6%). Additional categorization details and examples are available in Table 4.

## 3 EXPERIMENTS

We conducted experiments to evaluate the performance of mainstream Video Language Models (VLMs) on the testset of LongViTU, after instruction tuning on our training set. The outcomes illustrate that LongViTU poses distinct challenges to contemporary VLMs, regardless of their reliance on frame-based or sampling-based models. Moreover, instruction tuning with our training set improved performance across In-Distribution (ID) and several canonical Out-Of-Distribution (OOD) benchmarks, highlighting the exceptional generalization and robustness of LongViTU. This methodology expands the scope of conventional VQA datasets by encompassing a wider range of knowledge, a detailed analysis of the design principles underlying our dataset is also provided.

3.1 SETUP

**Settings and Baselines.** Following the methodology of LLava (Liu et al., 2024b), we utilized the full LongViTU training set to fine-tune various VLM models, evaluating their efficacy on the testset to tackle the novel challenges presented by our dataset in the context of long-form video understanding. The data formats were standardized, with inputs comprising relevant video clips for each task. Frame-based models such as mPLUG-OWL (Ye et al., 2023) and Video-LLaVA (Lin et al., 2023) uniformly extracted 8 frames from each input video. In comparison, models like LLaMA-VID (Li et al., 2023d) and Gemini-1.5-Pro (Reid et al., 2024) sampled one frame per second, whereas VideoAgent (Fan et al., 2024) opted for one frame every two seconds. These models generated answers to questions regarding the video content. To ensure fair comparison, a multi-level scoring criteria was designed. GPT-4 assessed the textual alignment between the generated answers and the ground truth, assigning scores based on predefined criteria and providing an average score for each subcategory.

**Metrics and Benchmarks.** In evaluating open-ended questions, traditional reliance on caption metrics is now considered inadequate. GPT-4 has demonstrated near-human performance in text comprehension and alignment. Therefore, we developed a multi-level scoring criteria, enabling GPT-4 to evaluate the correspondence between predicted answers and the ground truth, ensuring that the essential elements of the question are captured. Hallucinations or irrelevant responses result in a low score, whereas responses that accurately and concisely address key points receive a high score, details on the specific prompts are provided in Appendix B. Beyond testing on our dataset, we fine-tuned all models using the LongViTU training set and performed evaluations on ID benchmark EgoSchema (Mangalam et al., 2024) and OOD benchmarks VideoMME (Fu et al., 2024), WorldQA (Zhang et al., 2024b), and OpenEQA (Majumdar et al., 2024). These evaluations demonstrated enhanced performance compared to baseline models.

3.2 MAIN RESULT I: QUANTITATIVE ANALYSIS OF LONGVITU

The detailed quantitative evaluations of LongViTU are delineated in Table 2, from which we derive the following insights:

**Effective fine-tuning with LongViTU.** Upon employing LongViTU for training, all fine-tuned models exhibit enhanced performance on the testset, surpassing their initial zero-shot outcomes. Remarkably, the open-source model Video-LLaVa attains an average score of 50.7 post fine-tuning, approximating the leading commercial model Gemini-1.5-Pro at a dense sampling rate of 1 fps. This parity between top-tier open-source and commercial models underscores the persistent challenges posed by LongViTU in mastering long-form video content.

**Poor sampling models.** At a sampling frequency of 1 fps, the performance of LLaMA-VID is suboptimal compared to that of mainstream open-source video language models (VLMs), in both zero-shot and fine-tuned scenarios. This performance gap indicates a deficiency in the representational capabilities of existing dense sampling strategies, essential for effective long-form video analysis. LongViTU introduces the inaugural extensive, diverse, and high-quality dataset and benchmark, catering to this research domain.

**Analysis of blind QA.** Outcomes from pure text-based blind QA sessions are competitively robust, suggesting that using text as an intermediary may skew QA systems towards textual domain predictions. This enables a direct inference of questions from answers, we elaborate on the limitations of our pipeline in Appendix A.

**Spatial *vs*. temporal bias.** The noticeable underperformance in Spatiotemporal Understanding relative to Episodic Reasoning and Commonsense Inference accentuates prevalent issues. Tasks that require emphasis on spatial details prove exceptionally demanding within the scope of long-form video understanding, indicating unresolved complexities in this domain. Conversely, Episodic Memory Reasoning and Commonsense Inference derive benefits from the logical connectivity among sequential events, thus yielding superior results when leveraging text-based data.

3.3 MAIN RESULT II: QUANTITATIVE ANALYSIS ON BENCHMARKS

This section presents quantitative evaluations conducted on benchmarks that encompass In-Distribution (ID) and Out-of-Distribution (OOD) scenarios, specifically using the EgoSchema (Man-

Table 3: **Quantitative results on additional benchmarks.** The * denotes results obtained in a zero-shot manner, while ** indicates fine-tuned results following training on the LongViTU training set, △ *compared* highlighting the percentage difference in performance between their. Denote $s_2$ as the *stage 2* and $s_3$ as the *stage 3*, they are strictly following LLaMA-VID.

| Method | EgoSchema | VideoMME | | | | WorldQA | OpenEQA | | |
|---|---|---|---|---|---|---|---|---|---|
| | | Avg. | Short | Medium | Long | | Avg. | ScanNet | HM3D |
| VideoLLM-online | 47.4 | 13.7 | 24.3 | 16.7 | 0.0 | 30.0 | 23.3 | 24.8 | 20.4 |
| LLaMA-VID$^{*s_3}$ | 23.6 | 14.6 | 19.5 | 12.6 | 11.5 | 30.9 | 31.1 | 31.0 | 31.3 |
| LLaMA-VID$^{*s_2}$ | 30.4 | 16.7 | 22.6 | 15.3 | 12.2 | 32.0 | 31.9 | 31.8 | 32.1 |
| LLaMA-VID** | 34.0 | 17.2 | 23.8 | 15.4 | 12.2 | 32.2 | 33.6 | 33.5 | 33.8 |
| △ *compared* | +11.8% | +3.0% | +5.3% | +0.7% | +0.0% | +0.6% | +5.3% | +5.3% | +5.3% |
| Video-LLaVA* | 36.8 | 32.3 | 33.7 | 31.6 | 31.5 | 30.2 | 35.1 | 37.3 | 30.9 |
| Video-LLaVA** | 48.1 | 32.5 | 30.5 | 33.7 | 33.1 | 34.1 | 32.6 | 32.6 | 32.5 |
| △ *compared* | +30.7% | +0.6% | -9.5% | +6.6% | +5.1% | +12.9% | -7.1% | -12.6% | +5.2% |

galam et al., 2024), VideoMME (Fu et al., 2024), WorldQA (Zhang et al., 2024b), and OpenEQA (Majumdar et al., 2024) datasets as detailed in Table 3. Significant observations are summarized below:

**Failures of sampling models.** The LLaMA-VID (Li et al., 2023d) model showcased a notable decrement in performance during the zero-shot fashion of stage 3, which focuses on fine-tuning for long-for video, in contrast to its achievements in stage 2 comprising pre-training on images and brief video sequences. This performance gap reveals critical shortcomings in the strategy adopted for stage 3 of LLaMA-VID. We adjusted the finetuning strategy, with all finetuning on LLaMA-VID based on the stage 2, all results showed significant improvements over stage 2, with the highest being an 11.8% increase on EgoSchema.

**Better on longer videos.** Post fine-tuning, Video-LLaVA's performance declined mainly on shorter videos. Fine-tuning with LongViTU on longer videos also showed limited effectiveness on the VideoMME Short subset. The average video duration in OpenEQA is 49 seconds, which is shorter than the 83 seconds in the VideoMME Short subset, but much less than the Medium (563 seconds) and Long (2386 seconds) subsets. These results highlight the importance of LongViTU for improving understanding across different video durations.

**Challenges in long video processing.** The VideoLLM-online (Chen et al., 2024a) model demonstrated incoherent responses during the assessments of the Long subset of VideoMME, with no measurable predictions across the evaluation metrics. This underscores the significant challenges inherent in processing and understanding lengthy video content. The LongViTU dataset serves not only as a substantial foundation for pre-training but also as a crucial benchmark for evaluating the capabilities in long-form video comprehension.

## 3.4 QUALITATIVE EVALUATION

We present visualizations of various question-answering types in Figure 4 to facilitate a more thorough qualitative analysis.

**Spatial details.** As shown in Figure 4a, the dense distribution of numerous foreground objects within the scene led to incorrect zero-shot predictions from both Video-LLaVA and LLaMA-VID. After fine-tuning with LongViTU, the model effectively focused on finer spatial details, resulting in fully correct answers.

**Key moments.** In Figure 4b, Video-LLaVA successfully identified a key moment (a fleeting appearance of *"a plant on the windowsill"*), and provided a precise and concise response, which was awarded a perfect score of 100 by GPT-4. In contrast, LLaMA-VID, despite being fine-tuned, failed to capture sufficient details and received a score of 0.

**Temporal localization.** In Figure 4c, both Video-LLaVA and LLaMA-VID correctly identified the presence of *"two"* plug-in sockets in the kitchen at the end of a long video, providing accurate and succinct answers. Extracting such spatial information from extended video sequences poses a significant challenge, highlighting the effectiveness of LongViTU data in improving the generalization of long-form temporal localization.

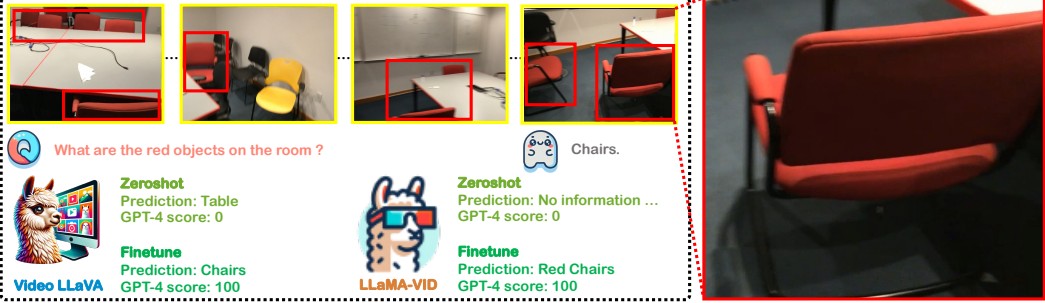

(a) **Spatial details.** Q: What are the red objects on the room ? A: Chairs.

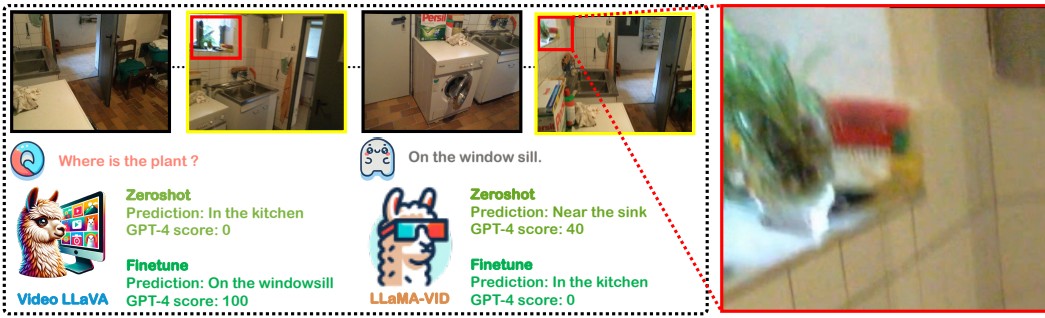

(b) **Key moments.** Q: Where is the plant ? A: On the window sill.

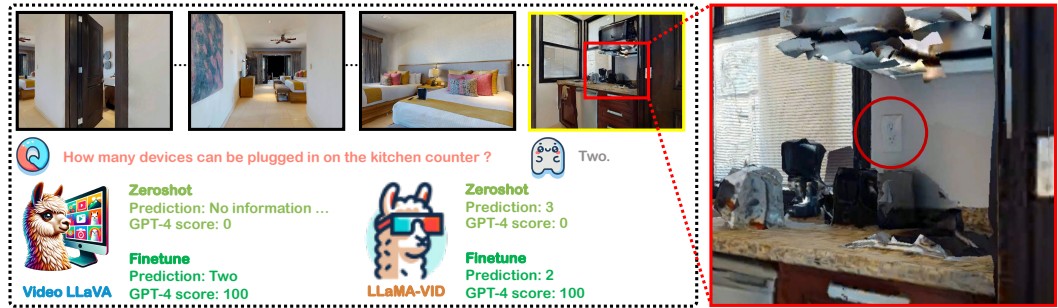

(c) **Temporal localization.** Q: How many devices can be plugged in on the kitchen counter ? A: Two.

Figure 4: **Qualitative results on LongViTU**. The yellow box indicates the key frame containing the answer, while the red box highlights relevant objects.

## 3.5 FUTURE RESEARCH

Our distinct and rational design endows LongViTU with numerous advantageous properties. Beyond serving as a pre-training dataset and evaluation benchmark for long-form video understanding, thereby enhancing generalization to OOD benchmarks, it also contributes groundbreaking insights into streaming video processing and the exploration of video memory storage mechanisms in end-to-end models. Future research will extend beyond long-form video understanding to integrate streaming video processing and question-answering, fostering a more comprehensive approach.

## 4 RELATED WORK

**Large language models.** Large language models (LLMs), such as InstructGPT (Ouyang et al., 2022), GPT-4 (Achiam et al., 2023), LLaMA (Touvron et al., 2023a), and LLaMA-2 (Touvron et al., 2023b), have demonstrated notable capabilities in text processing, which have motivated their use in generating large-scale multimodal datasets. These models convert different modalities into structured textual descriptions, which can then be used to prompt GPT-4 to produce multimodal content. This process effectively uses text as a bridge to unify various data forms, enabling new approaches in dataset automation.

**Instruction tuning dataset.** LLaVA (Liu et al., 2024b) was among the first to leverage foundational vision models to generate image captions and detect bounding boxes, which were subsequently processed by ChatGPT or GPT-4 to create QA tasks based on images. Building on this, methods like Bongard-OpenWorld (Wu et al., 2023), Video-LLaVA (Lin et al., 2023), and VideoChat (Li et al., 2023c) extended these principles to video data, transitioning from individual image QA to video-based QA. By sampling multiple frames from videos and applying LLaVA's procedure, these approaches generate video QA datasets using structured frame descriptions, object categories, and attributes. However, this basic repetition across frames, combined with the input length limits of LLMs, constrains the number of frames that can be analyzed, thereby reducing the comprehensiveness of the resulting datasets.

**Long-context language models.** Even the most advanced long-context LLMs, such as GPT-4, ChatGLM (GLM et al., 2024), Baichuan2 (Yang et al., 2023a), and InternLM2 (Cai et al., 2024b), capable of handling input sequences beyond 128k tokens, experience substantial performance degradation when confronted with long, intricate texts. They struggle to manage the redundancy and disorder inherent in detailed descriptions of numerous video frames, limiting their ability to generate effective video QA. Unlike static images, videos inherently require an understanding of temporal dynamics, making event correlation crucial for video comprehension. The current frame-based extension approach fails to address this temporal aspect adequately, often resulting in QA generation that lacks depth beyond individual frame analysis.

**Long-form video understanding.** Instruction tuning based on the LLaVA paradigm (Liu et al., 2024b) has demonstrated strong potential for multimodal alignment and understanding, including tasks like captioning and visual question-answering (Brown, 2020; Anil et al., 2023; Team et al., 2023; Li et al., 2023b; Dai et al., 2023; Yang et al., 2024; Alayrac et al., 2022). While these methods perform effectively for individual images and short videos, extending to long-form video understanding presents significant challenges (Song et al., 2024; Lin et al., 2023; Maaz et al., 2023; Zhang et al., 2023; Wang et al., 2024; Zhang et al., 2024a). This difficulty primarily arises due to the vast number of visual tokens produced by visual encoders, ranging from 576 to 2880 tokens per image in the case of LLaVA-NeXT (Liu et al., 2024a). As the number of frames increases, the context-window length of LLMs is quickly exceeded. Recent methods have attempted to reduce the number of visual tokens via resamplers that connect visual encoders to LLMs (Li et al., 2023b;d; Cai et al., 2024a; Cheng et al., 2024), but this often compromises the quality of visual representation, leading to suboptimal outcomes. More refined techniques for pruning or merging visual features could provide a promising direction to address these limitations (Chen et al., 2024b; Shang et al., 2024; Jin et al., 2024; Zhou et al., 2024).

## 5  CONCLUSION

We present LongViTU, a large-scale dataset for long-form video understanding, incorporating video memory and explicit timestamp annotations. Our approach organizes video content hierarchically into a tree structure to tackle the complexities of generating QA datasets for extended video content. Using a sliding window mechanism, we capture both temporal and spatial context, ensuring that QA pairs align well with video content and cover diverse and informative aspects. A self-evaluation and revision process further improves QA quality by reducing hallucinations, redundancy, and irrelevant content. Fine-tuning on the LongViTU training set led to significant performance improvements on both LongViTU and other benchmarks, demonstrating its efficacy and generalizability. Future work will explore memory storage strategies using explicit timestamps to further enhance long-form video understanding and streaming video QA.

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

## A  LIMITATION STATEMENT

We have designed LongViTU, a pretraining dataset and evaluation benchmark for long video understanding, presenting significant challenges to current Video Language Models. Neither open-source nor commercial models have effectively addressed this problem yet, and we continue to explore potential solutions. In this paper, we present the challenge of long-form video understanding and memory-based streaming video QA, focusing on how to efficiently integrate detailed visual features in extended video content. We also highlight the current limitations of existing methods, our automated pipeline is adaptable to various scenarios. Although self-evaluation and revision improve data quality, some generated data, while not incorrect, may lack meaningful context, potentially affecting the model. To address this, we propose refining the pipeline and adding manual inspection to better align the benchmark with real-world human understanding and QA. This benchmark will help evaluate progress and performance in this field, and the work is ongoing.

## B  MORE DETAILS ON BUILDING LONGViTU

### B.1  HIERARCHICAL VIDEO TREE CONSTRUCTION

This subsection outlines the hierarchical video tree construction process, with details provided in Section 2.1.1. Each stage's corresponding prompts are described in the following sections.

---

**Algorithm 1** Hierarchical Video Tree Construction

---

**Input:** Annotated video frames $Frames$, events $Events$, segments $Segments$
**Output:** Hierarchical video tree $\mathcal{T}_{\text{video}}$
1: $\mathcal{T}_{\text{video}} = \emptyset$
2: **for** $s = 1$ to $len(Segments)$ **do**
3:  $segment = Segments[s]$
4:  $d_s = \emptyset; t_s^s = \emptyset; t_e^s = \emptyset$
5:  **for** $e = 1$ to $len(segment.events)$ **do**
6:   $event = segment.events[e]$
7:   $d_e = GPT4(\langle d_f^H \rangle_{f=1}^F)$
8:   $t_s^e = event.start$
9:   $t_e^e = event.end$
10:   $segment.events[e] = \langle d_e, t_s^e, t_e^e \rangle$
11:  **end for**
12:  $d_s = GPT4(\{segment.events\})$
13:  $t_s^s = segment.start$
14:  $t_e^s = segment.end$
15:  $\mathcal{T}_{\text{video}} = \mathcal{T}_{\text{video}} \cup \{\langle d_s, t_s^s, t_e^s, segment.events \rangle\}$
16: **end for**
17: **return** $\mathcal{T}_{\text{video}}$;

---

**Frame level.** We utilize the `internlm-xcomposer2-vl-7b-4bit` model with the following prompt for dense captioning at the frame level:

> <ImageHere>Identify each object in the image, describe their positions, and detail their appearance.

**Event level.** We employ the ChatCompletion API of the `gpt-4-turbo` model with the following prompt to refine event level descriptions:

> Write a very concise narrative in one sentence, including visual details from "Frames" that depict an "Event", do not use any unrelated information.
>
> "Event" describes an action in a video, with "C" representing me and other letters like 'X' and 'Y' standing for different people, transform these for a smoother narrative.
>
> "Frames" show detailed visuals and space details of objects in each moment during the "Event".
>
> Event: {event}
>
> Frames: {frames}
>
> Just return narrative that summarizes the episodic memory depicted in this video, only focuses on spatial details and temporal correlations.
>
> Narrative:

**Segment level.** We utilize the ChatCompletion API of the `gpt-4-turbo` model to generate segment level descriptions:

> Integrate sequential event descriptions of video content into a very concise summary in one sentence, from my perspective for a smoother narrative. Each segment should capture a sequence of closely related actions, events, or scenes. Using "index" to represent the start and end of each segment, do not use any unrelated information.
>
> Step-by-step:
> 1. Review event descriptions and group consecutive events that are closely related into a segment.
> 2. For each group of events, write a brief summary.
>
> "index" represents order of event, "event" outlines this moment.
>
> Video Content:
> {video content}
>
> Return each segment in JSON format: "start": start index, "end": end index, "segment": brief description of video segment. Assemble all segments into a single Python list, ensuring output is neatly organized and strictly adheres to this JSON format.
>
> Segments:

### B.2 LONG-FORM QA GENERATION

We utilize the ChatCompletion API of the `gpt-4-turbo` model to generate QA pairs on the selected subtree:

Task:
Construct episodic memory of video content through question-answer pairs that encapsulate spatial and temporal aspects within selected events.

Step-by-Step Instructions:
1. Selection of Events: Select either a single specific event or a series of interrelated events from the video content ('Memory Content'). For each selected event or sequence of events, generate question-answer pairs that reflect their spatial and temporal characteristics. Use "index" to designate the chronological order of these memory events.
2. Creation of Question-Answer Pairs: From the selected events, formulate questions that will be posed later in the video related to a single, specific event ('Ask Content'). These pairs should mimic a retrospective dialogue between me and an AI assistant, where I pose questions and the AI provides answers based on the video content. Reference events and segments to make dialogue more naturally narrative, avoiding direct references "index" or timestamps.
3. Categorization of Questions: Categorize each question under a specific type such as: Object, Attribute, Location, Action, Function, Affordance, Comparison, Relationship, Causality, Motivation, Planning, Risk, or any other category you suggest.

Output Format:
Return question-answer pairs in JSON format: "memory": [list of memory events index], "ask": event index where question is posed, "type": question type, "question": question, "answer": answer. Assemble all pairs into a single Python list, ensuring the output is neatly organized and strictly adheres to this JSON format.

Term Definitions of Video Content:
- segment: a brief summary covering a sequence of related events.
- events: multiple related events within a segment.
- index: sequential position of an event within the overall video content.
- event: spatial-temporal details associated with each moment in the video.

Memory Content:
{memory content}

Ask Content:
{ask content}

Question-Answer pairs:

## B.3 SELF-REVISION

We utilize the ChatCompletion API of the `gpt-4-turbo` model to perform self-revision:

Please review and correct the following question-answer pair about video content. Simplify the question-answer pair to directly represent the core information without redundant details, ensuring the question is natural and concise, and the answer is direct and clear.

Identify the correct type of the QA pair: Object, Attribute, Location, Action, Function, Affordance, Comparison, Relationship, Causality, Motivation, Planning, Risk, or Other. Do not add or fabricate content. Remove redundant event numbers and express the event directly.

Original QA:
{original qa}

Return the Revised QA as a dict:
{"revised type": revised QA type, "revised question": revised question, "revised answer": revised answer}

Revised QA:

## B.4 EVALUATION METRICS

We use the `internlm-xcomposer2-vl-7b-4bit` model to perform evaluation by designed scoring criteria:

As a scoring expert, your responsibility is to evaluate the accuracy of a model's response to a specific question about video content. You will be provided with the 'question' asked about the video, the 'answer' which is the correct answer based on the video, and the 'prediction' which is the model's response. Your task is to assess how accurately the model's 'prediction' answers the 'question' in relation to the 'answer'.

Question:
{question}

Answer:
{answer}

Prediction:
{prediction}

Scoring Criteria:
Level 1: The 'prediction' is unrelated to the 'question' or unintelligible, containing significant errors or irrelevant characters. Score: 0.
Level 2: The 'prediction' is completely off-topic, not reflecting the factual content of the 'answer'. Score: 20.
Level 3: The 'prediction' somewhat response the 'question' but includes errors or irrelevant details not found in the 'answer'. Score: 40.
Level 4: The 'prediction' generally response the 'question' but has some inaccuracies or irrelevant details compared to the 'answer'. Score: 60.
Level 5: The 'prediction' accurately response the 'question' and is mostly consistent with the 'answer', with only minor discrepancies. Score: 80.
Level 6: The 'prediction' perfectly response the 'question' and fully aligns with the facts provided in the 'answer'. Score: 100.

Only provide the numerical score based on the criteria above without any additional commentary.

Score:

## B.5 MORE LONGVITU EXAMPLES

Table 4: **Examples of each category in LongViTU.** We demonstrate the ratio of each concept category and more examples of LongViTU. For a more intuitive perspective, you may refer to Figure 3.

| QA Category | Ratio | Question Example |
|---|---|---|
| Object | 12.2% | What am I holding in my hand? 
 What items are on the table? 
 What is the object on the ground? |
| Attribute | 10.7% | What is the color of that clothing? 
 What is the material of the cup? 
 What is the shape of this table? |
| Location | 15.5% | Where am I in the house right now? 
 Where is the key placed in? 
 Is that woman by the window? |
| Action | 16.6% | What is that man doing? 
 What am I doing by the counter? 
 What did he do after he came out of the house? |
| Transition | 8.1% | Where did he go after leaving here? 
 What change happened to that cup? 
 What just appeared on the ground? |
| Interaction | 3.4% | Which hand did I use to pick up this wrench? 
 What did I take out of the microwave? 
 Am I pushing the bike or riding it? |
| Causality | 5.4% | What happened after I pressed that button? 
 What happened after I opened the box? 
 What made it move? |
| Motivation | 7.5% | Why should I leave the room? 
 Why does she want to open the cabinet? 
 Why is he crying? |
| Planning | 5.4% | How do I get to the backyard? 
 How do I repair this house? 
 How do I get the tool? |
| Risk | 2.7% | What dangers does that saw pose? 
 What dangers are there in the kitchen? 
 What dangers are nearby I am driving? |
| Function | 6.4% | What is the function of this tool? 
 What is the function of this box? 
 What is the function of this knife? |
| Affordance | 4.6% | What can this stone be used for? 
 What can this glass bottle be used for? 
 What can this cloth be used for? |

