# OpenReview forum: "LongViTU: Instruction Tuning for Long-Form Video Understanding"
_ICLR.cc/2025/Conference — ICLR 2025 Conference Withdrawn Submission_

### Official Review · Reviewer_sDiL · 2024-11-01

**Soundness:** 3
**Presentation:** 2
**Contribution:** 2
**Rating:** 6
**Confidence:** 4

**Summary:**

In this work, the authors propose a LongViTU benchmark for Long-Form Video Understanding. Basically, they leverage Ego4D as data source, and develop a three-stage pipeline for QA annotation and revision. First, it builds up a hierarchical video tree to describe videos in different temporal scales. Second, they apply a sliding window approach to any subtree, and generate QA of subtree by GPT4. Third, they use GPT-4 to make a thorough revision of the generated question-answering pairs.

**Strengths:**

1 Topic is good. Long-form video understanding is a challenging but important problem. Developping a benchmark for instruct tuning and evaluation is critical in this problem.

2 Experiments are sufficient. The experimental studies are interesting to show the challenges and potentials of this benchmark.

**Weaknesses:**

1 This benchmark is based on EGO4D. Hence, the annotation would be similar to EgoTaskQA. As shown in Table 1, the difference is the increasing scale of data set and the newly-added timestep annotations. Is such timestep annotation important or not? Are there any expermental results to show its impact on your benchmark ?

2 The hierarchical video tree style design is similar to [MoVQA: A Benchmark of Versatile Question-Answering for Long-Form Movie Understanding, arXiv:2312.04817].

3 The paper writing should be refined. The structure is OK, while the content is not quite easy to read.

**Questions:**

Please see weakness section.

---

### Official Review · Reviewer_fVx1 · 2024-11-03

**Soundness:** 2
**Presentation:** 3
**Contribution:** 2
**Rating:** 5
**Confidence:** 4

**Summary:**

The paper introduces LongViTU, a large-scale dataset designed for long-form video understanding, featuring approximately 121k question-answer pairs across 900 hours of video content. It addresses challenges in long-form video understanding by offering a dataset with diverse real-world scenarios, explicit timestamp labels, long certificate lengths, fine-grained categorization, and open-ended precise QA pairs. LongViTU is curated to facilitate instruction tuning for long-form videos, involving the organization of video content into a hierarchical tree and incorporating self-revision mechanisms to ensure high-quality QA pairs. The authors primarily validate the effectiveness of LongViTU through experiments conducted on two different models.

**Strengths:**

1.	The approach of organizing video content into a hierarchical tree structure is innovative. This method allows for the generation of question-answer pairs that capture both spatial and temporal details, which is a creative extension of existing video understanding frameworks.
2.	The dataset provides fine-grained categorization of questions, which is crucial for advancing the understanding of complex video content and adds depth to the quality of the dataset.

**Weaknesses:**

1.	In Table 2, it can be observed that there is a lack of differentiation in the benchmark. The performance gap between the best-performing Gemini-1.5-Pro and the other models is not evident. According to the reviewer, in most existing benchmarks, Gemini-1.5-Pro demonstrates a significant performance advantage over Video-LLaVA.

2.	The proposed benchmark employs GPT-4 for assessment, which may introduce additional bias.

3.	The validation method employed was released some time ago, and its baseline performance is no longer highly competitive compared to more recent models. It remains unclear whether it can still deliver significant performance improvements on more recently proposed models.

**Questions:**

1.	Is it possible that using GPT-4 for evaluation may struggle to distinguish fine-grained semantics? For instance, if sentences differ by only one or two keywords but convey significantly different meanings, how would GPT-4 rate them in such cases?

2.	Can LongViTU still deliver substantial performance improvements on models that perform better?

---

### Official Review · Reviewer_rfFt · 2024-11-03

**Soundness:** 3
**Presentation:** 2
**Contribution:** 2
**Rating:** 5
**Confidence:** 4

**Summary:**

This paper introduces LongViTU, a novel large-scale dataset (~121k QA pairs, ~900h videos) for long-form video understanding.  The authors address the limitations of existing video question-answering (VQA) datasets by focusing on several key aspects:  diverse real-world scenarios (leveraging Ego4D), explicit timestamp labels for QA-related events, long average certificate length (4.6 minutes), fine-grained categorization of QA pairs (spatiotemporal understanding, episodic reasoning, commonsense inference), and open-ended, precise QA generation.  A hierarchical pipeline, employing LLMs (primarily GPT-4) at multiple stages (hierarchical video tree construction, long-form QA generation, self-revision), is used for automatic dataset creation.  Experiments demonstrate the challenges posed by LongViTU to existing video language models (VLMs), showing a performance gap even between open-source and commercial models. Fine-tuning on LongViTU improves performance on both in-distribution and out-of-distribution benchmarks.

**Strengths:**

1. LongViTU explicitly addresses the limitations of temporal context, length, and fine-grained question types from the perspective of sft.  The hierarchical pipeline for automatic dataset generation is a sound procedure to create long-form annotations from bottom to top. Its sheer scale of the dataset (~900 hours of video) and its diversity in terms of scenarios and question types are decent.  The use of Ego4D ensures real-world relevance.
2. The paper includes a thorough quantitative evaluation on LongViTU and several benchmark datasets, demonstrating the effectiveness of the dataset and highlighting the challenges it presents.  The use of GPT-4 for scoring is a reasonable approach given the open-ended nature of the QA pairs.  Qualitative examples further illustrate the dataset's capabilities. The availability of the dataset, fine-tuned models, and code is a valuable contribution to the community.

**Weaknesses:**

1. The reliance on LLMs (GPT-4) throughout the pipeline raises concerns about potential biases inherited from the pre-training data of these models.  Moreover, a hierarchical pipeline may cause error cumulation, making the bias even worse. A thorough analysis of potential biases in the generated QA pairs is missing.
2. While self-revision is employed, a more robust human evaluation of the dataset quality would strengthen the paper's claims.  The current human evaluation seems limited to Appendix B.
3. Experiments need improvements. The number of models evaluated in the benchmark is too limited, and some of the current long video large language models, such as LongVA, LongVILA, have not been included in the evaluation. The model performance used to validate the training dataset's effectiveness is too weak (for instance, LLama-VID performs below random chance on VideoMME), and the improvements achieved after fine-tuning are relatively minor.

**Questions:**

1. How were the specific parameters for the sliding window (five segments) determined? What is the sensitivity of the results to changes in this parameter?
2. What is the inter-annotator agreement (IAA) for the human annotations used in the Ego4D dataset, and how does this affect the quality of LongViTU?
3. What are the computational costs associated with generating and processing LongViTU?
4. Can you provide a more detailed analysis of the biases present in the generated QA pairs?
5. How does the performance of the fine-tuned models change with different sizes of the LongViTU training set?

---

### Official Review · Reviewer_KRZ6 · 2024-11-04

**Soundness:** 3
**Presentation:** 2
**Contribution:** 2
**Rating:** 3
**Confidence:** 4

**Summary:**

The paper introduces LongViTU for video understanding, which comprises approximately 121k question-answer pairs across 900 hours of video content, focusing on long-context videos that require rich knowledge and reasoning. The authors propose a hierarchical pipeline for generating high-quality QA pairs with explicit timestamp labels, catering to diverse real-world scenarios. LongViTU is curated to support fine-grained and open-ended QA. The paper also presents experiments demonstrating the performance gap between open-source and commercial models on this benchmark and the effectiveness of SFT on LongViTU.

**Strengths:**

- The paper is easy to follow, and the experiments are clearly described.
- The dataset is of high quality, featuring a large number of QA pairs and encompassing a variety of diverse scenarios.

**Weaknesses:**

- Figure 1: The icons, while visually appealing, come across as unprofessional and occupy space that could be better utilized to present more information.
- Ablation Studies: The paper lacks ablation studies for different-level captions. For instance, it would be beneficial to know if event-level captions can be skipped without significant detriment.
- Results: Additional results are necessary to clarify the performance of different Multi-modal Large Language Models (MLLMs) on LongViTU videos with varying durations.
- Comparison with ShareGPT4Video[1]: The authors of ShareGPT4Video present a progressive framework that generates detailed captions for diverse videos. In contrast, LongViTU focuses solely on ego-centric videos due to its dependence on human annotation, which potentially limits its application and robustness for general QA, as evidenced in Table 3.

---
Reference:

[1] Chen, Lin et al. “ShareGPT4Video: Improving Video Understanding and Generation with Better Captions.” ArXiv abs/2406.04325 (2024): n. pag.

**Questions:**

See weaknesses.

---

### Note · Authors · 2024-11-13

I have read and agree with the venue's withdrawal policy on behalf of myself and my co-authors.